# Gradient Descent for Spiking Neural Networks

**Dongsung Huh**
Salk Institute
La Jolla, CA 92037
huh@salk.edu

**Terrence J. Sejnowski**
Salk Institute
La Jolla, CA 92037
terry@salk.edu

## Abstract

Most large-scale network models use neurons with static nonlinearities that produce analog output, despite the fact that information processing in the brain is predominantly carried out by dynamic neurons that produce discrete pulses called spikes. Research in spike-based computation has been impeded by the lack of efficient supervised learning algorithm for spiking neural networks. Here, we present a gradient descent method for optimizing spiking network models by introducing a differentiable formulation of spiking dynamics and deriving the exact gradient calculation. For demonstration, we trained recurrent spiking networks on two dynamic tasks: one that requires optimizing fast ($\approx$ millisecond) spike-based interactions for efficient encoding of information, and a delayed-memory task over extended duration ($\approx$ second). The results show that the gradient descent approach indeed optimizes networks dynamics on the time scale of individual spikes as well as on behavioral time scales. In conclusion, our method yields a general purpose supervised learning algorithm for spiking neural networks, which can facilitate further investigations on spike-based computations.

## 1 Introduction

The brain operates in a highly decentralized event-driven manner, processing multiple asynchronous streams of sensory-motor data in real-time. The main currency of neural computation is spikes: *i.e.* brief impulse signals transmitted between neurons. Experimental evidence shows that brain's architecture utilizes not only the rate, but the precise timing of spikes to process information [1].

Deep-learning models solve simplified problems by assuming *static units* that produce analog output, which describes the time-averaged firing-rate response of a neuron. These *rate-based* artificial neural networks (ANNs) are easily differentiated, and therefore can be efficiently trained using gradient descent learning rules. The recent success of deep learning demonstrates the computational potential of trainable, hierarchical distributed architectures.

This brings up the natural question: What types of computation would be possible if we could train spiking neural networks (SNNs)? The set of implementable functions by SNNs subsumes that of ANNs, since a spiking neuron reduces to a rate-based unit in the high firing-rate limit. Moreover, in the low firing-rate range in which the brain operates (1$\sim$10 Hz), spike-times can be utilized as an additional dimension for computation. However, such computational potential has never been explored due to the lack of general learning algorithms for SNNs.

### 1.1 Prior work

Dynamical systems are most generally described by ordinary differential equations, but linear time-invariant systems can also be characterized by impulse response kernels. Most SNN models are constructed using the latter approach, by defining a neuron's membrane voltage $v_i(t)$ as a weighted

linear summation of kernels $K_{ij}(t - t_k)$ that describe how the spike-event of neuron $j$ at previous time $t_k$ affects neuron $i$ at time $t$. When the neuron's voltage approaches a sufficient level, it generates a spike in deterministic [2, 3, 4, 5, 6, 7, 8, 9, 10, 11, 12] or stochastic manner [13, 14, 15, 16, 17]. These kernel-based neuron models are known as spike response models (SRMs).

The appeal of SRMs is that they can simulate SNN dynamics without explicit integration steps. However, this representation takes individual spike-times as the state variables of SNNs, which causes problems for learning algorithms when spikes are needed to be created or deleted during the learning process. For example, Spikeprop [2] and its variants [3, 4, 5, 6, 18] calculate the derivatives of spike-times to derive accurate gradient-based update rules, but they are only applicable to problems where each neuron is constrained to generating a predefined number of spikes.

Currently, learning algorithms compatible with variable spike counts have multiple shortcomings: Most gradient-based methods can only train "visible neurons" that directly receive desired target output patterns [7, 8, 9, 11, 13, 17]. While extensions have been proposed to enable training of hidden neurons in multilayer [10, 14, 16, 19] and recurrent networks [15], they require neglecting the derivative of the self-kernel terms, *i.e.* $K_{ii}(t)$, which is crucial for the gradient information to propagate through spike events. Moreover, the learning rules derived for specific neuron dynamics models cannot be easily generalized to other neuron models. Also, most methods require the training data to be prepared in spike-time representations. For instance, they use loss functions that penalize the difference between the desired and the actual output spike-time patterns. In practice, however, such spike-time data are rarely available.

Alternative approaches take inspiration from biological spike-time dependent plasticity (STDP) [20, 21], and reward-modulated STDP process [22, 23, 24]. However, it is generally hard to guarantee convergence of these bottom-up approaches, which do not consider the complex effects network dynamics nor the task information in designing of the learning rule.

Lastly, there are rate-based learning approaches, which convert trained ANN models into spiking models [25, 26, 27, 28, 29, 30], or apply rate-based learning rules to training SNNs [31]. However, these approaches can at best replicate the solutions from rate-based ANN models, rather than exploring computational solutions that can utilize spike-times.

## 1.2 New learning framework for spiking neural networks

Here, we derive a novel learning approach for training SNNs represented by ordinary differential equations. The state vector is composed of dynamic variables, such as membrane voltage and synaptic current, rather than spike-time history. This approach is compatible with the usual setup in optimal control, which allows gradient calculation by using the existing tools in optimal control. Moreover, resulting process closely resembles the familiar backpropagation rule, which can fully utilize the existing statistical optimization methods in deep learning framework.

Note that, unlike the prior literature, our work here provides not just a single learning rule for a particular model and task, but a general framework for calculating gradient for arbitrary network architecture, neuron models, and loss functions. Moreover, the goal of this research is not necessarily to replicate a biological learning phenomenon, but to derive efficient learning methods that can explore the computational solutions implementable by the networks of spiking neurons in biology. The trained SNN model could then be analyzed to reveal the computational processes of the brain, or provide algorithmic solutions that can be implemented with neuromorphic hardwares.

## 2 Methods

### 2.1 Differentiable synapse model

In spiking networks, transmission of neural activity is mediated by synaptic current. Most models describe the synaptic current dynamics as a linear filter process which instantly activates when the presynaptic membrane voltage $v$ crosses a threshold: *e.g.*,

$$\tau \dot{s} = -s + \sum_k \delta(t - t_k). \tag{1}$$

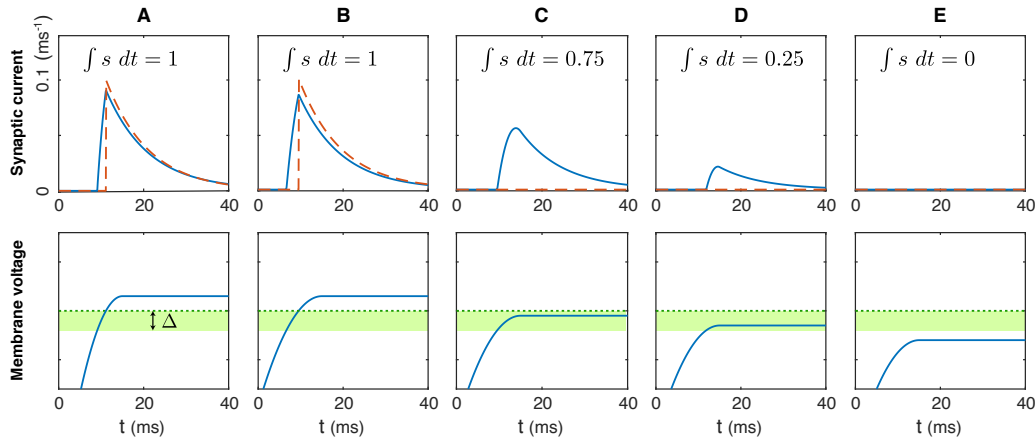

Figure 1: Differentiability of synaptic current dynamics: The synaptic current traces from eq (2) (solid lines, upper panels) are shown with the corresponding membrane voltage traces (lower panels). Here, the gate function is $g = 1/\Delta$ within the active zone of width $\Delta$ (shaded area, lower panels); $g = 0$ otherwise. (A,B) The pre-synaptic membrane voltage depolarizes beyond the active zone. Despite the different rates of depolarization, both events incur the same amount of charge in the synaptic activity: $\int s\,dt = 1$. (C,D,E) Graded synaptic activity due to insufficient depolarization levels that do not exceed the active zone. The threshold-triggered synaptic dynamics in eq (1) is also shown for comparison (red dashed lines, upper panels). The effect of voltage reset is ignored for the purpose of illustration. $\tau = 10$ ms.

where $\delta(\cdot)$ is the Dirac-delta function, and $t_k$ denotes the time of $k^{\text{th}}$ threshold-crossing. Such threshold-triggered dynamics generates discrete, all-or-none responses of synaptic current, which is non-differentiable.

Here, we replace the threshold with a gate function $g(v)$: a non-negative ($g \geq 0$), unit integral ($\int g\,dv = 1$) function with narrow support[1], which we call the active zone. This allows the synaptic current to be activated in a gradual manner throughout the active zone. The corresponding synaptic current dynamics is

$$\tau \dot{s} = -s + g\dot{v}, \tag{2}$$

where $\dot{v}$ is the time derivative of the pre-synaptic membrane voltage. The $\dot{v}$ term is required for the dimensional consistency between eq (1) and (2): The $g\dot{v}$ term has the same $[\text{time}]^{-1}$ dimension as the Dirac-delta impulses of eq (1), since the gate function has the dimension $[\text{voltage}]^{-1}$ and $\dot{v}$ has the dimension $[\text{voltage}][\text{time}]^{-1}$. Hence, the time integral of synaptic current, *i.e.* charge, is a dimensionless quantity. Consequently, a depolarization event beyond the active zone induces a constant amount of total charge regardless of the time scale of depolarization, since

$$\int s\,dt = \int g\dot{v}\,dt = \int g\,dv = 1.$$

Therefore, eq (2) generalizes the threshold-triggered synapse model while preserving the fundamental property of spiking neurons: *i.e.* all supra-threshold depolarizations induce the same amount of synaptic responses regardless of the depolarization rate (Figure 1A,B). Depolarizations below the active zone induce no synaptic responses (Figure 1E), and depolarizations within the active zone induce graded responses (Figure 1C,D). This contrasts with the threshold-triggered synaptic dynamics, which causes abrupt, non-differentiable change of response at the threshold (Figure 1, dashed lines).

Note that the $g\dot{v}$ term reduces to the Dirac-delta impulses in the zero-width limit of the active zone, which reduces eq (2) back to the threshold-triggered synapse model eq (1).

The gate function, without the $\dot{v}$ term, was previously used as a differentiable model of synaptic connection [32]. In such a model, however, a spike event delivers varying amount of charge depending on the depolarization rate: the slower the presynaptic depolarization, the greater the amount of charge delivered to the post-synaptic targets.

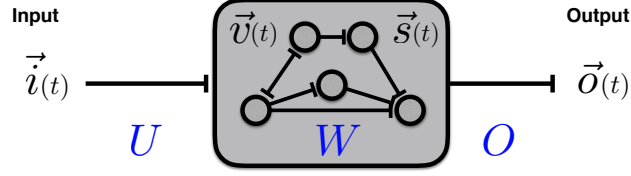

Figure 2: The model receives time varying input, $\vec{i}(t)$, processes it through a network of spiking neurons, and produces time varying output, $\vec{o}(t)$. The internal state variables are the membrane voltage $\vec{v}(t)$ and the synaptic current $\vec{s}(t)$.

## 2.2 Network model

To complete the input-output dynamics of a spiking neuron, the synaptic current dynamics must be coupled with the presynaptic neuron's internal state dynamics. For simplicity, we consider differentiable neural dynamics that depend only on the the membrane voltage and the input current:

$$\dot{v} = f(v, I). \tag{3}$$

The dynamics of an interconnected network of neurons can then be constructed by linking the dynamics of individual neurons and synapses eq (2,3) through the input current vector:

$$\vec{I} = W\vec{s} + U\vec{i} + \vec{I}_o, \tag{4}$$

where $W$ is the recurrent connectivity weight matrix, $U$ is the input weight matrix, $\vec{i}(t)$ is the input signal for the network, and $\vec{I}_o$ is the tonic current. Note that this formulation describes general, fully connected networks; specific network structures can be imposed by constraining the connectivity: *e.g.* triangular matrix structure $W$ for multi-layer feedforward networks.

Lastly, we define the output of the network as the linear readout of the synaptic current:

$$\vec{o}(t) = O\vec{s}(t),$$

where $O$ is the readout matrix. The overall schematic of the model is shown in Figure 2.

All of the network parameters $W, U, O, \vec{I}_o$ can be tuned to minimize the total cost, $C \equiv \int l(t)\, dt$, where $l$ is the cost function that evaluates the performance of network output for given task.

## 2.3 Gradient calculation

The above spiking neural network model can be optimized via gradient descent. In general, the exact gradient of a dynamical system can be calculated using either Pontryagin's minimum principle [33], also known as backpropagation through time, or real-time recurrent learning, which yield identical results. We present the former approach here, which scales better with network size, $\mathcal{O}(N^2)$ instead of $\mathcal{O}(N^3)$, but the latter approach can also be straightforwardly implemented.

Backpropagation through time for the spiking dynamics eq (2,3) utilizes the following backpropagating dynamics of adjoint state variables $(p_v, p_s$. See Supplementary Materials):

$$-\dot{p}_v = \partial_v f\, p_v - g\dot{p}_s \tag{5}$$
$$-\tau \dot{p}_s = -p_s + \xi, \tag{6}$$

where $p_v, p_s$ are the *modified* adjoints of $v$ and $s$, $\partial_v f \equiv \partial f/\partial v$, and $\xi$ is called the *error current*. For the recurrently connected network eq (4), the error current vector has the following expression

$$\vec{\xi} = W^{\intercal}(\partial_I \vec{f}\, p_v) + \vec{\partial}_s l, \tag{7}$$

which links the backpropagating dynamics eq (5,6) of individual neurons. Here, $\partial_I f \equiv \partial f/\partial I$, $(\partial_I f\, p_v)_k \equiv (\partial f/\partial I)_k p_{v_k}$, and $(\partial_s l)_k \equiv \partial l/\partial s_k$.

Interestingly, the coupling term of the backpropagating dynamics, $g\dot{p}_s$, has the same form as the coupling term $g\dot{v}$ of the forward-propagating dynamics. Thus, the same gating mechanism that

mediates the spiked-based communication of signals also controls the propagation of error in the same sparse, compressed manner.

Given the adjoint state vectors that satisfy eq (5,6,7), the gradient of the total cost with respect to the network parameters can be calculated as

$$\nabla_W C = \int (\partial_I \vec{f} \, p_v) \, \vec{s}^\mathsf{T} \, dt$$

$$\nabla_U C = \int (\partial_I \vec{f} \, p_v) \, \vec{i}^\mathsf{T} \, dt$$

$$\nabla_{I_o} C = \int (\partial_I \vec{f} \, p_v) \, dt$$

$$\nabla_O C = \int \vec{\partial}_o l \, \vec{s}^\mathsf{T} \, dt$$

where $(\partial_o l)_k \equiv \partial l / \partial o_k$. Note that the gradient calculation procedure involves multiplication between the presynaptic input source and the postsynaptic adjoint state $p_v$, which is driven by the $g \dot{p}_s$ term: *i.e.* the product of postsynaptic spike activity and temporal difference of error. This is analogous to reward-modulated spike-time dependent plasticity (STDP) [24].

## 3  Results

We demonstrate our method by training spiking networks on dynamic tasks that require information processing over time. Tasks are defined by the relationship between time-varying input-output signals, which are used as training examples. We draw mini-batches of $\approx 50$ training examples from the signal distribution, calculate the gradient of the average total cost, and use stochastic gradient descent [34] for optimization.

Here, we use a cost function $l$ that penalizes the readout error and the overall synaptic activity:

$$l = \frac{\|\vec{o} - \vec{o}_d\|^2 + \lambda \|\vec{s}\|^2}{2},$$

where $\vec{o}_d(t)$ is the desired output, and $\lambda$ is a regularization parameter.

### 3.1  Predictive Coding Task

We first consider *predictive coding* tasks [35, 36], which optimize spike-based representations to accurately reproduce the input-ouput behavior of a linear dynamical system of full-rank input and output matrices. Analytical solutions for this class of problems can be obtained in the form of non-leaky integrate and fire (NIF) neural networks, although insignificant amount of leak current is often added [36]. The solutions also require the networks to be equipped with a set of instantaneous synapses for fast time-scale interactions between neurons, as well slower synapses for readout. Despite its simplicity, the predictive coding framework reproduces important features of biological neural networks, such as the balance of excitatory and inhibitory inputs and efficient coding [35]. Also, its analytical solutions provide a great benchmark for assessing the effectiveness of our learning method.

The membrane voltage dynamics of a NIF neuron is given by

$$f(v, I) = I.$$

Here, we impose two thresholds at $v_{\theta+} = 1$ and $v_{\theta-} = -1$, and the reset voltage at $v_{\text{reset}} = 0$, where the $v_{\theta-}$ threshold would trigger negative synaptic responses. This bi-threshold NIF model naturally fits with the inherent sign symmetry of the task, and also provides an easy solution to ensure that the membrane voltage stays within a finite range. However, the training also works with the usual single threshold model. We also introduce two different synaptic time constants, as proposed in [35, 36]: a fast constant $\tau = 1$ ms for synapses for the recurrent connections, and a slow constant $\tau_s = 10$ ms for readout.

In the predictive-coding task, the desired output signal is the low-pass filtered version of the input signal:

$$\tau_s \dot{\vec{o}}_d = -\vec{o}_d + \vec{i},$$

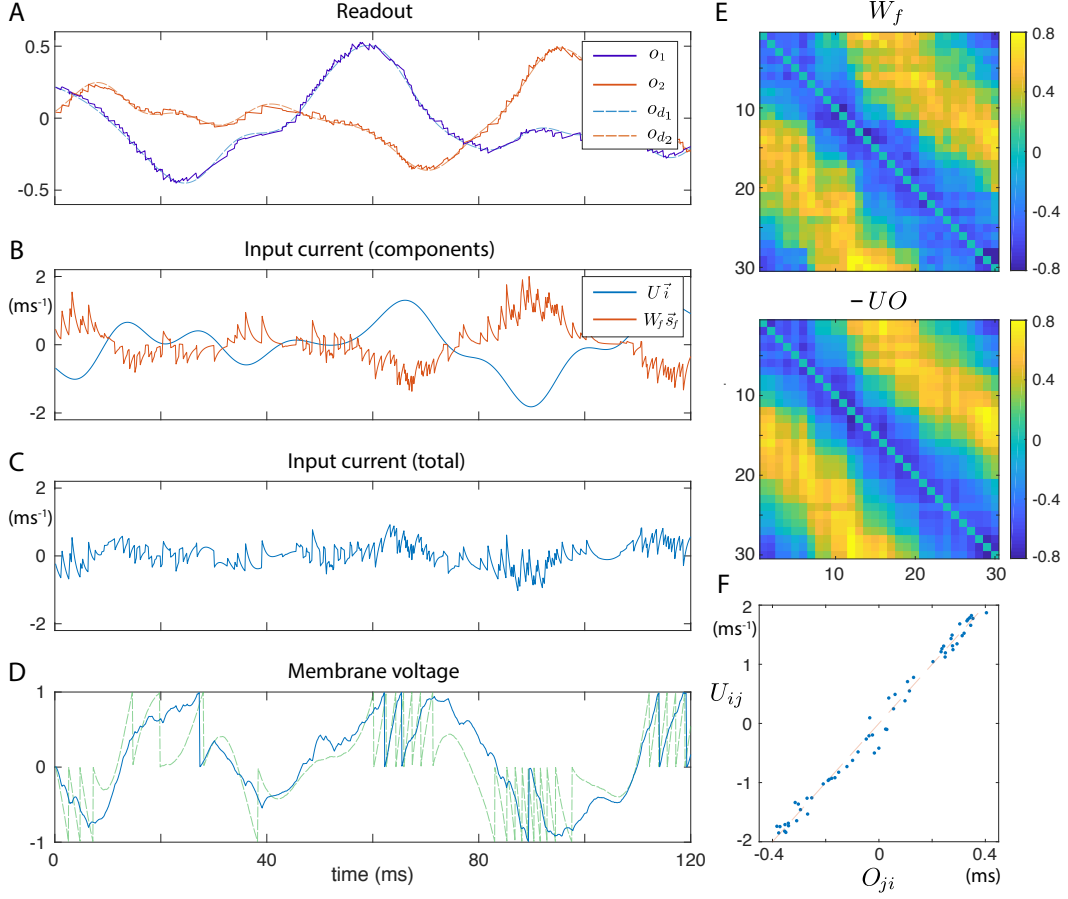

Figure 3: Balanced dynamics of a spiking network trained for auto-encoding task. (A) Readout signals: actual (solid), and desired (dashed). (B) Input current components into a single neuron: external input current ($U\vec{i}(t)$, blue), and fast reccurent synaptic current ($W_f \vec{s}_f(t)$, red). (C) Total input current into a single neuron ($U\vec{i}(t) + W_f \vec{s}_f(t)$). (D) Single neuron membrane voltage traces: the actual voltage trace driven by both external input and fast reccurent synaptic current (solid, 6 spikes), and a virtual trace driven by external input only (dashed, 29 spikes). (E) Fast recurrent weight: trained ($W_f$, above) and predicted ($-UO$, below). Diagonal elements are set to zero to avoid self-excitation/inhibition. (F) Readout weight $O$ vs input weight $U$.

where $\tau_s$ is the slow synaptic time constant [35, 36]. The goal is to accurately represent the analog signals using least number of spikes. We used a network of 30 NIF neurons, 2 input and 2 output channels. Randomly generated sum-of-sinusoid signals with period 1200 ms were used as the input.

The output of the trained network accurately tracks the desired output (Figure 3A). Analysis of the simulation reveals that the network operates in a tightly balanced regime: The fast recurrent synaptic input, $W\vec{s}(t)$, provides opposing current that mostly cancels the input current from the external signal, $U\vec{i}(t)$, such that the neuron generates a greatly reduced number of spike outputs (Figure 3B,C,D). The network structure also shows close agreement to the prediction. The optimal input weight matrix is equal to the transpose of the readout matrix (up to a scale factor), $U \propto O^\mathsf{T}$, and the optimal fast recurrent weight is approximately the product of the input and readout weights, $W \approx -UO$, which are in close agreement with [35, 36, 37]. Such network structures have been shown to maintain tight input balance and remove redundant spikes to encode the signals in most efficient manner: The representation error scales as $1/K$, where $K$ is the number of involved spikes, compared to the $1/\sqrt{K}$ error of encoding with independent Poisson spikes.

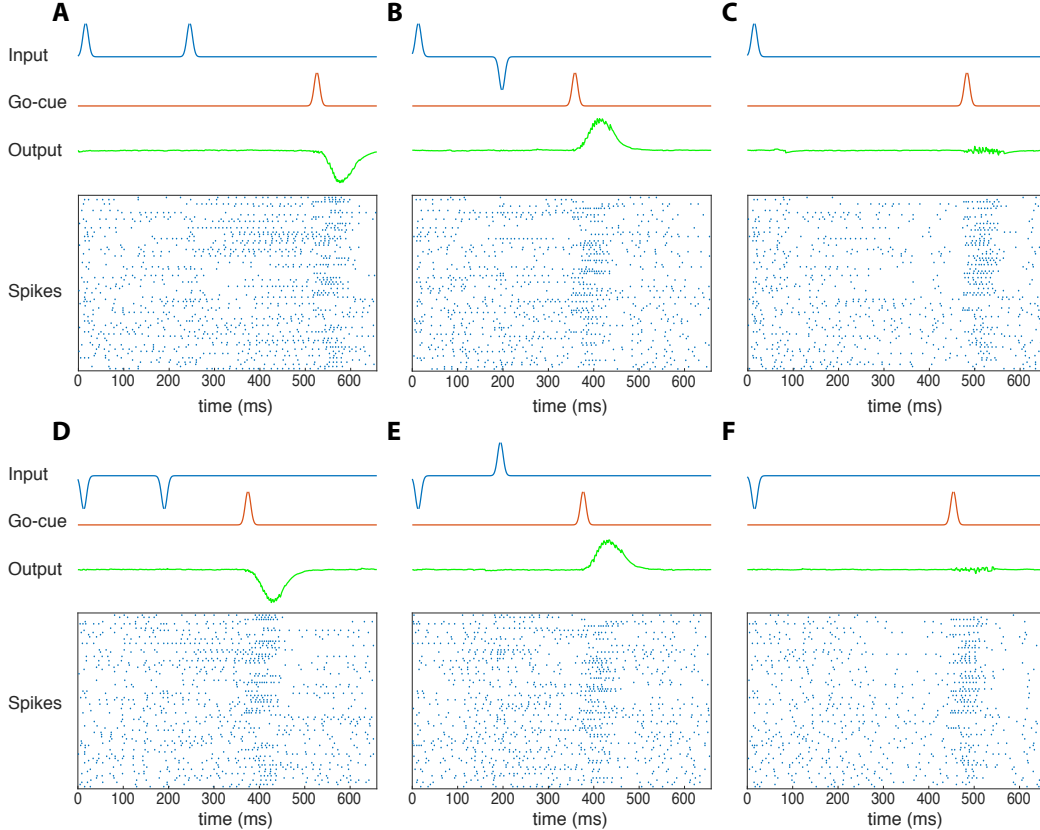

Figure 4: Delayed-memory XOR task: Each panel shows the single-trial input, go-cue, output traces, and spike raster of an optimized QIF neural network. The y-axis of the raster plot is the neuron ID. Note the similarity of the initial portion of spike patterns for trials of the same first input pulses (A,B,C vs D,E,F). In contrast, the spike patterns after the go-cue signal are similar for trials of the same desired output pulses: (A,D: negative output), (B,E: positive output), and (C,F: null output).

## 3.2 Delayed-memory XOR task

A major challenge for spike-based computation is in bridging the wide divergence between the time-scales of spikes and behavior: How do millisecond spikes perform behaviorally relevant computations on the order of seconds?

Here, we consider a *delayed-memory XOR* task, which performs the *exclusive-or* (XOR) operation on the input history stored over extended duration. Specifically, the network receives binary pulse signals, $+$ or $-$, through an input channel and a go-cue through another channel. If the network receives two input pulses since the last go-cue signal, it should generate the XOR output pulse on the next go-cue: *i.e.* a positive output pulse if the input pulses are of opposite signs ($+-$ or $-+$), and a negative output pulse if the input pulses are of equal signs ($++$ or $--$). Additionally, it should generate a null output if only one input pulse is received since the last go-cue signal. Variable time delays are introduced between the input pulses and the go-cues.

A simpler version of the task was proposed in [26], whose solution involved first training an analog, rate-based ANN model and converting the trained ANN dynamics with a larger network of spiking neurons ($\approx 3000$), using the results from predictive coding [35]. It also required a dendritic nonlinearity function to match the transfer function of rate neurons.

We trained a network of 80 quadratic integrate and fire (QIF) neurons[2], whose dynamics is

$$f(v, I) = (1 + \cos(2\pi v))/\tau_v + (1 - \cos(2\pi v))I,$$

[2]NIF networks fail to learn the delayed-memory XOR task: the memory requirement for past input history drives the training toward strong recurrent connections and runaway excitation.

also known as Theta neuron model [38], with the threshold and the reset voltage at $v_\theta = 1$, $v_{\text{reset}} = 0$. Time constants of $\tau_v = 25$, $\tau_f = 5$, and $\tau = 20$ ms were used, whereas the time-scale of the task was $\approx 500$ ms, much longer than the time constants. The intrinsic nonlinearity of the QIF spiking dynamics proves to be sufficient for solving this task without requiring extra dendritic nonlinearity. The trained network successfully solves the delayed-memory XOR task (Figure 4): The spike patterns exhibit time-varying, but sustained activities that maintain the input history, generate the correct outputs when triggered by the go-cue signal, and then return to the background activity. More analysis is needed to understand the exact underlying computational mechanism.

This result shows that out algorithm can indeed optimize spiking networks to perform nonlinear computations over extended time.

## 4 Discussion

We have presented a novel, differentiable formulation of spiking neural networks and derived the gradient calculation for supervised learning. Unlike previous learning methods, our method optimizes the spiking network dynamics for general supervised tasks on the time scale of individual spikes as well as the behavioral time scales.

Exact gradient-based learning methods, such as ours, may depart from the known biological learning mechanisms. Nonetheless, these methods provide a solid theoretical foundation for understanding the principles underlying biological learning rules. For example, our result shows that the gradient update occurs in a sparsely compressed manner near spike times, similar to reward-modulated STDP, which depends only on a narrow 20 ms window around the postsynaptic spike. Further analysis may reveal that certain aspects of the gradient calculation can be approximated in a biologically plausible manner without significantly compromising the efficiency of optimization. For example, it was recently shown that the biologically implausible aspects of backpropagation method can be resolved through feedback alignment in rate-based multilayer feedforward networks [39]. Such approximations could also apply to spiking neural networks.

Here, we coupled the synaptic current model with differentiable single-state spiking neuron models. We want to emphasize that the synapse model can be coupled to any neuron model, including biologically realistic multi-state neuron models with action potential dynamics [3], including the Hodgkin-Huxley model, the Morris-Lecar model and the FitzHugh-Nagumo model; and an even wider range of neuron models with internal adaptation variables and neuron models having non-differentiable reset dynamics, such as the leaky integrate and fire model, the exponential integrate and fire model, and the Izhikevich model. This will be examined in the future work.

## Footnotes

[1]Support of a function $g : X \to \mathbb{R}$ is the subset of the domain $X$ where $g(x)$ is non-zero.

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

## Supplementary Materials: Gradient calculation for the spiking neural network

**Pontryagin's minimum principle**   According to [33], the Hamiltonian for the dynamics eq (2,3,4) is

$$\mathcal{H} = \sum_i \bar{p}_{v_i} \dot{v}_i + \bar{p}_{s_i} \dot{s}_i + l(\vec{s})$$

$$= \sum_i (\bar{p}_{v_i} + g_i \bar{p}_{s_i}/\tau) f_i \; - \bar{p}_{s_i} s_i/\tau + l(\vec{s}),$$

where $\bar{p}_{v_i}$ and $\bar{p}_{s_i}$ are the adjoint state variables for the membrane voltage $v_i$ and the synaptic current $s_i$ of neuron $i$, respectively, and $l(\vec{s})$ is the cost function. The back-propagating dynamics of the adjoint state variables are:

$$-\dot{\bar{p}}_{v_i} = \frac{\partial \mathcal{H}}{\partial v_i} = (\bar{p}_{v_i} + g_i \bar{p}_{s_i}/\tau)\partial_v f_i \; + f_i g_i' \bar{p}_{s_i}/\tau$$

$$-\dot{\bar{p}}_{s_i} = \frac{\partial \mathcal{H}}{\partial s_i} = \sum_j (\bar{p}_{v_j} + g_j \bar{p}_{s_j}/\tau) \cdot \partial_I f_j \; W_{ji} - \bar{p}_{s_i}/\tau + l_{s_i}$$

where $f_i \equiv f(v_i, I_i)$, $g_i \equiv g(v_i)$, $\partial_v f_i \equiv \partial f/\partial v_i$, $\partial_I f_i \equiv \partial f/\partial I_i$, $g_i' \equiv dg/dv_i$, and $l_{s_i} \equiv \partial l/\partial s_i$.

This formulation can be simplified via change of variables, $p_{v_i} \equiv \bar{p}_{v_i} + g \bar{p}_{s_i}/\tau$, $p_{s_i} \equiv \bar{p}_{s_i}/\tau$, which yields

$$\mathcal{H} = \vec{p}_v \cdot \vec{f} - \vec{p}_s \cdot \vec{s} + l$$

$$-\dot{p}_{v_i} = \partial_v f_i \; p_{v_i} - g_i \dot{p}_{s_i}$$

$$-\tau \dot{p}_{s_i} = -p_{s_i} + l_{s_i} + \sum_j W_{ji} \partial_I f_j \; p_{v_j},$$

where we used $\dot{p}_{v_i} = \dot{\bar{p}}_{v_i} + f_i \; g_i' \bar{p}_{s_i}/\tau + g_i \dot{\bar{p}}_{s_i}/\tau$.

The gradient of the total cost can be obtained by integrating the partial derivative of the Hamiltonian with respect to the parameter (*e.g.* $\partial \mathcal{H}/\partial W_{ij}$, $\partial \mathcal{H}/\partial U_{ij}$, $\partial \mathcal{H}/\partial I_{o_i}$, $\partial \mathcal{H}/\partial O_{ij}$).

