[Reviews · NeurIPS 2018]

Reviewer 1



The work presents a novel approach to introduce gradient descent to optimise spiking neural networks. The main contribution is to replace the Dirac-Delta function in the activation function of spiking neural networks with a differentiable gate function allowing for a gradient descent optimisation to be used. The approach is reasonable and has the potential to overcome major challenges in the wider application of spiking neural networks. The paper is, however, not very easy to follow. The postsynaptic adjoint state p_v is not defined in eq 5,6 and a lot of assumptions have been made on the detailed knowledge of the reader. The test cases presented are standard in the literature and do not demonstrate the potential of the new approach.

Reviewer 2



The authors present interesting work on gradient descent calculation for spiking neural networks, a topic which has seen vigorous research in the last couple of years. The work is timely, and could be significantly improved by including a computation analysis of the trained networks, a pursuit that is described as left for future research. Major Comments: Much recent work has focused on developing semi-biologically plausible rules for training spiking nets. Therefore, it would be useful for the authors to elaborate on how their method is more appropriate. For example, the authors state that some of these papers ”…neglect a derivative term that is crucial for computing accurate gradient flow through spike events” (lines 55-56). It would be useful to elaborate on these specific claims. The authors make reference to Supplementary Materials for gradient calculation, but there are no Supplementary Materials? Can the method be extended to training addition parameters of the network, such as the various time constants of each neuron model? There is no reference to Figure 2 at all, although the reference should be in section 4.2. Section 5.1 does not contain any specific results. I feel like there is a Figure missing somewhere. References to Figure 3 are incorrectly labeled as references to Figure 2. Minor Comments: - Line 67: typo, “ANN models purple” - Line 75: typo, “resembles to”

Reviewer 3



This paper introduces a smooth thresholding technique which enables practically standard gradient descent optimization to be applied to spiking neural networks. Since the spiking threshold is usually set at a certain membrane potential, the function "spike or no spike" is a function of voltage whose distributional derivative is a dirac Delta at the threshold. By replacing this Dirac delta by a finite positive function g(v) with tight support around the threshold, and which integrates to 1, the step function "spike or no spike" is replaced by a function that increases continuously from 0 to 1 across the support of g. In turn, this setup can be placed into standard differential equation models governing spikes, while retaining the possibility of having meaningful gradient signal for parameter optimization. Two experiments are evaluated, an autoencoding task and a delayed-memory-XOR task, which are both shown to be trainable with the proposed setup. This is a good contribution which removes some previously required restrictions for training spiking neuron models (which are enumerated in the introduction). An analysis that is missing and which imposes itself quite naturally is the behavior of the training as a function of support size/smoothness of g. One could for example evaluate g_a(v) = a g(a(v - v0) + v0) with g starting as very smooth and becoming more pointy as a increases. At some point training should become impossible due to convergence of this method to standard thresholding (+ numerical issues). Next, one could also evaluate making this function continuously sharper during training.